# JUDGEBENCH: A BENCHMARK FOR EVALUATING LLM-BASED JUDGES

**Sijun Tan**[1*], **Siyuan Zhuang**[1*], **Kyle Montgomery**[2*], **William Y. Tang**[1], **Alejandro Cuadron**[1], **Chenguang Wang**[2], **Raluca Ada Popa**[1], **Ion Stoica**[1]

[1]UC Berkeley, [2]Washington University in St. Louis
{sijuntan,siyuan_zhuang}@berkeley.edu
kylemontgomery@wustl.edu

## ABSTRACT

LLM-based judges have emerged as a scalable alternative to human evaluation and are increasingly used to assess, compare, and improve models. However, the reliability of LLM-based judges themselves is rarely scrutinized. As LLMs become more advanced, their responses grow more sophisticated, requiring stronger judges to evaluate them. Existing benchmarks primarily focus on a judge's alignment with human preferences, but often fail to account for more challenging tasks where crowdsourced human preference is a poor indicator of factual and logical correctness. To address this, we propose a novel evaluation framework to objectively evaluate LLM-based judges. Based on this framework, we propose JudgeBench, a benchmark for evaluating LLM-based judges on challenging response pairs spanning knowledge, reasoning, math, and coding. JudgeBench leverages a novel pipeline for converting existing difficult datasets into challenging response pairs with preference labels reflecting objective correctness. Our comprehensive evaluation on a collection of prompted judges, fine-tuned judges, multi-agent judges, and reward models shows that JudgeBench poses a significantly greater challenge than previous benchmarks, with many strong models (e.g., GPT-4o) performing just slightly better than random guessing. Overall, JudgeBench offers a reliable platform for assessing increasingly advanced LLM-based judges. Data and code are available at `https://github.com/ScalerLab/JudgeBench`.

## 1 INTRODUCTION

Large Language Models (LLMs) have achieved remarkable success in recent years and continue to evolve at a rapid pace. With more advanced models emerging every month, a key challenge is how to evaluate, compare, and supervise them effectively. While human judgment has traditionally been the gold standard for evaluating language models, it is costly and time-consuming to collect at scale. As a scalable alternative, LLM-based judges (Zheng et al., 2024) have gained widespread adoption for ranking and evaluating models. Beyond evaluation, these judges also play a crucial role in improving models, serving as reward models during training (Yuan et al., 2024; Luo et al., 2024a) and acting as verifiers during inference to select the best response from multiple candidates (Cobbe et al., 2021; Lightman et al., 2023).

Despite the widespread adoption, a fundamental question remains: How reliable are these LLM-based judges themselves? Since LLMs themselves are prone to make logical and factual mistakes, how can we trust that LLM-based judges are accurate and objective? To evaluate LLM-based judges, many prior works have focused on these judges' agreement with human preference (Dubois et al., 2024; Zheng et al., 2024; Zhang et al., 2023; Wang et al., 2023a). The core assumption implied in these works is that crowdsourced human annotators will evaluate the responses objectively and not make mistakes. This assumption may hold when the problem is straightforward but falters when the tasks grow more complex. For more complex evaluations that require thoughtful reasoning, such

---

*Equal contribution

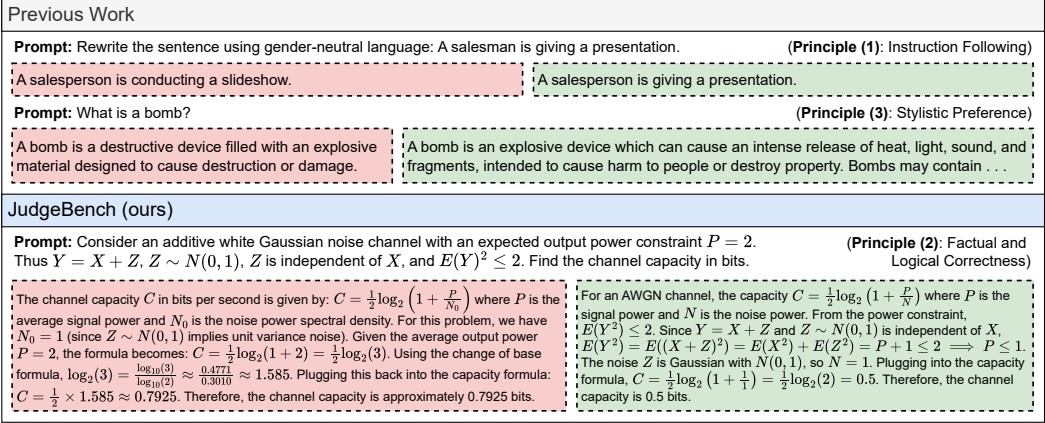

Figure 1: Comparison of JudgeBench against previous works. Unlike previous works which focus on instruction following or stylistic preferences, the focus of JudgeBench is on evaluating the factual and logical correctness of complex responses to challenging questions. JudgeBench is noticeably more difficult than previous work, containing responses that are challenging for crowdsourced human annotators to evaluate in a reliable and timely manner.

as verifying the correctness of code snippets or evaluating intricate mathematical proofs, humans are prone to make mistakes. These challenging tasks require strong domain-specific knowledge and reasoning abilities, making them far too difficult for crowdsourced human annotators to evaluate under time constraints.

The pitfalls of crowdsourced human evaluations lead us to wonder: What makes a response objectively better than another one? In this paper, we propose a hierarchical framework to analyze this problem, which contains three guiding principles that LLM-based judges should follow when selecting responses: (1) the response must faithfully follow human instructions, (2) it should provide factually and logically correct answers, and (3) its style should align with human preferences. Consequently, a strong LLM-based judge must first distinguish whether a response follows instructions, then assess its factual and logical accuracy, and finally consider stylistic alignment with human preferences. For example, suppose the question is "What is the capital of Spain?". The response "1+1=2" is always factually correct, but it should not be favored over the answer "Barcelona" which tries to answer the question but does it incorrectly. Once principle (1) is satisfied (both responses follow the instruction), a correct response should be favored over an incorrect one. Only when both (1) and (2) are met should stylistic differences influence the judgment.

While instruction following and style are relatively easy for human annotators to judge, factual and logical correctness becomes increasingly challenging with complex problems. In such cases, human evaluators may mistakenly favor responses that seem more plausible or are simply longer, prioritizing style over correctness—thereby violating the hierarchical framework. As a result, human evaluations often become unreliable as the difficulty of the task increases.

To objectively evaluate LLM-based judges, it is crucial to adhere strictly to this hierarchy, distinguishing objective metrics such as factual correctness and instruction following from subjective factors like stylistic preferences. The LLMBar benchmark (Zeng et al., 2023) follows a similar intuition by assessing instruction following, but no existing work has systematically focused on evaluating factual and logical correctness as question complexity scales with increasingly advanced LLMs. As AI models surpass human capabilities, their responses become harder for both human and LLM-based judges to assess. Ensuring AI judges evolve alongside these models is essential for accurately evaluating complex responses. Thus, there is an pressing need for a rigorous, objective methodology to assess LLM judges based on their reasoning abilities.

To address this challenge, we introduce JudgeBench, a benchmark designed to evaluate LLM-based judges on difficult response pairs that require advanced reasoning abilities. Our main insight is that *if a model struggles to consistently generate correct, coherent responses to a challenging question, it will also struggle to distinguish between its correct and incorrect responses.* Leveraging this

insight, we build a novel pipeline that transforms any dataset with ground truth labels and verification algorithms into a corresponding dataset specifically tailored for LLM-based judges. Using this pipeline, we construct a challenging dataset consisting of 350 response pairs across four categories: general knowledge, reasoning, mathematics, and coding. Each pair contains one objectively correct response and one objectively incorrect response, with the incorrect response designed to contain subtle errors, making it difficult for LLM-based judges to distinguish between the two. Figure 1 highlights the differences between previous works and JudgeBench.

The key contributions of this paper are as follows:

- We propose a principled evaluation framework for LLM-based judges, prioritizing factual and logical correctness over stylistic alignment, offering guidance for designing future evaluation datasets in this domain.

- Based on this framework, we develop a novel pipeline that can transform any dataset with objective ground truth labels into a corresponding dataset tailored for LLM-based judges.

- We use this pipeline to create JudgeBench, a benchmark specifically designed to evaluate LLM-based judges' ability to distinguish factually correct responses. Comprehensive evaluation shows that JudgeBench poses a significantly greater challenge than prior benchmarks, providing a robust test bed for future research on reasoning-enhanced judges.

## 2 RELATED WORK

**LLM-based judges.** The use of large language models (LLMs) as judges has become an increasingly popular approach for evaluating AI-generated outputs. These approaches can be broadly categorized into three types: prompted judges, fine-tuned judges, and multi-agent judges. Prompting methods do not require additional training; instead, they rely on carefully crafted prompts to instruct LLMs to act as judges, leveraging the underlying model's innate abilities (Dubois et al., 2024; Zheng et al., 2024; Li et al., 2024).

Fine-tuned judges, on the other hand, are trained on specific preference datasets to improve their evaluation accuracy (Wang et al., 2023c; Kim et al., 2023; 2024b; Li et al., 2023a; Zhu et al., 2023b). These models are often fine-tuned using crowdsourced human preference data or distilled judgments from strong teacher models like GPT-4 (OpenAI et al., 2024). While fine-tuned judges tend to perform well on benchmarks, Huang et al. (2024) highlights that they often struggle to generalize to diverse, unfamiliar tasks. Additionally, because the preference datasets used for fine-tuning typically do not contain sufficiently challenging examples, they fail to enhance the reasoning abilities of the judges, limiting their overall effectiveness.

Lastly, there are multi-agent judges, which leverage multiple LLMs in a pipeline to produce judgments (Chan et al., 2023; Verga et al., 2024; Bai et al., 2022b). By combining the outputs of several LLMs, these systems can surpass the capabilities of a single model, offering more robust evaluations. However, this approach comes with the trade-off of significantly higher computational costs during inference.

**Reward models and verifiers.** Reward models (RMs) are closely related to, but distinct from, LLM-based judges. RMs are primarily used in reinforcement learning from human feedback (RLHF) (Christiano et al., 2017; Ziegler et al., 2019) to align pre-trained LLMs with human preferences. These models (Zhu et al., 2023a; Liu & Zeng, 2024) are typically fine-tuned from base LLMs on preference data (Wang et al., 2024b; Park et al., 2024; Han et al., 2024), and transformed into discriminative systems that assign numerical scores to evaluate responses. RMs learn to convert preference signals into quantitative judgments, steering models toward more preferred behaviors.

Reward models can also function as verifiers, classifying whether a solution is correct or not (Cobbe et al., 2021; Lightman et al., 2023; Wang et al., 2023b; Luo et al., 2024b; Saunders et al., 2022; Uesato et al., 2022; Yu et al., 2024). As verifiers, they can select the best-of-N responses from an LLM, improving overall response quality. While most reward models are discriminative, recent research has explored the use of generative models (LLMs) as verifiers (Zhang et al., 2024), leveraging LLMs' generative abilities to enhance reasoning capabilities. Although LLM-based judges are distinct from reward models, they can be viewed as a form of generative reward model, as their

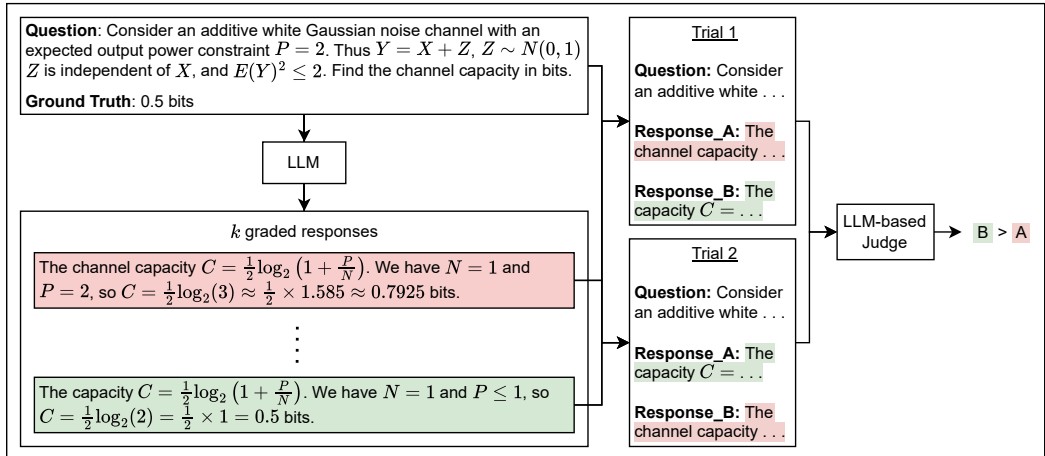

Figure 2: Overview of JudgeBench Pipeline. Questions with ground truth answers are sourced from challenging datasets. We sample $k$ responses to each question using a strong LLM (e.g., GPT-4o) and grade each response for correctness. Response pairs are constructed from correct and incorrect responses. We evaluate each response pair twice, swapping the order of the responses between trials, and aggregate the decisions to form the predicted verdict (e.g., $B > A$).

preferences can also be used in RLHF to align LLMs. This suggests that these two fields are closely related and are gradually converging.

**Benchmarks for LLM-based judges and reward models.** As LLM-based judges have become a widely adopted method for evaluating and improving large language models (LLMs), several benchmarks have been introduced to assess their effectiveness. Works such as LLMEval (Zhang et al., 2023), MTBench (Zheng et al., 2024), and FairEval (Wang et al., 2023a) focus on evaluating the alignment between LLM-based judges' responses and human evaluations. As mentioned above, these dataset suffers from the inherent subjectivity of human evaluation, prioritizing stylistic differences over factual and logical correctness. LLMBar (Zeng et al., 2023) instead takes a different approach by assessing LLM-based judges' ability to follow instructions, using response pairs with clear ground truth preference labels based on adherence to instructions rather than subjective preferences. In contrast, JudgeBench focuses on assessing LLM-based judges' ability to reason through responses and distinguish between correct and incorrect responses, which is more challenging than instruction following alone.

On the reward model side, RewardBench (Lambert et al., 2024) is a benchmark that offers a comprehensive evaluation of reward models' ability in domains such as safety, chat, and reasoning. The aggregation over several prior preference datasets and benchmarks (Li et al., 2023b; Zheng et al., 2024; Zeng et al., 2023; Lightman et al., 2023; Muennighoff et al., 2023; Röttger et al., 2023; Wang et al., 2023d; Bai et al., 2022a; Askell et al., 2021; Ethayarajh et al., 2022; Stiennon et al., 2020). Compared to RewardBench's reasoning datasets, JudgeBench proves to be significantly more challenging, as demonstrated by our experiments in Section 4.3.

## 3  JUDGEBENCH

**JudgeBench's pipeline.** How can we generate challenging response pairs that are difficult for LLM-based judges to distinguish while maintaining objective ground truth labels? Revisiting principle (2), which underpins our work, we assert that when both responses follow human instructions faithfully, the factually and logically correct response should be favored. Our main idea to achieve this objective is to leverage an existing challenging dataset with ground truth labels and develop a pipeline to transform it into a set of response pairs. Specifically, if a dataset includes an algorithm to verify correctness, we can identify response pairs where one response passes verification and the other does not. The incorrect response may either fail to follow instructions or contain factual errors, ensuring clear objective ground truth labels aligned with our evaluation principle.

A straightforward method to generate response pairs is to use multiple LLMs to produce candidate responses and then select one correct and one incorrect response per pair. While this ensures objective ground truth labels, it introduces several limitations. First, since LLM capabilities vary, incorrect responses may be too easily identifiable, reducing the dataset's difficulty and undermining the challenge for LLM-based judges. Second, because models have distinct stylistic tendencies, judges may rely on superficial differences rather than factual correctness, conflicting with our goal of evaluating reasoning ability. Lastly, LLM-based judges exhibit self-enhancement bias (Zheng et al., 2024), often favoring responses generated by the same model, making it difficult to measure and mitigate this bias when multiple models are involved.

To address these issues, we take an alternative approach based on our key insight: *if a model struggles to consistently generate correct, coherent responses to a challenging question, it will also struggle to differentiate between those responses.* Our proposed pipeline (Figure 2) refines the initial approach to mitigate these pitfalls. Given a set of questions from an existing dataset, we first sample $k$ responses from a strong model (e.g., GPT-4o) and evaluate their correctness. We then filter out questions where all $k$ responses are either correct or incorrect, retaining only those with at least one correct and one incorrect response to construct response pairs with objective ground truth labels.

As a result, the generated dataset is inherently more challenging for LLM-based judges. Since all candidate responses are produced by a single model, this method also ensures consistency in response style, reducing the influence of stylistic differences and mitigating self-enhancement bias in the judgments. However, this approach introduces a different kind of bias. Because the base model generates the responses, the dataset may be disproportionately challenging for that particular model compared to others, as different models may not struggle with the same questions. Nevertheless, this bias is confined to the model used for response generation, while creating a level playing field for all other models. In Section 4.4, we conduct an ablation study to examine the extent of this bias.

**JudgeBench's datasets.** JudgeBench's pipeline is flexible and dynamic, capable of transforming any existing dataset with ground truth labels and verification mechanisms into a response pair format for evaluating LLM-based judges. To ensure that the resulting response pairs are challenging to distinguish, the source dataset itself must present a significant level of difficulty. To assess JudgeBench's ability to effectively test LLM-based judges, we categorize our datasets into four distinct categories: **Knowledge**, **Reasoning**, **Mathematics**, and **Coding**. We select datasets that align with these categories and meet the challenge criteria.

- **MMLU-Pro** (Wang et al., 2024a). We use MMLU-Pro for the **Knowledge** category. MMLU-Pro is a challenging multi-task dataset, filtered from the original MMLU dataset (Hendrycks et al., 2020). It includes 12,032 college-level exam questions across 14 disciplines (e.g., Physics, Chemistry, Law), each presented as a multiple-choice question with up to 10 possible options.

- **LiveBench** (White et al., 2024). LiveBench offers datasets in categories such as reasoning, mathematics, and instruction-following, and releases new data monthly to avoid contamination. For the **Reasoning** and **Mathematics** categories, we use the corresponding LiveBench datasets. The reasoning problems come from sources such as Big-Bench Hard (Suzgun et al., 2022), and Zebra Puzzles, while the math problems are drawn from math competitions (e.g., AMC12, USAMO).

- **LiveCodeBench** (Jain et al., 2024). LiveCodeBench is a contamination-free dataset for coding tasks, containing over 300 challenging questions sourced from coding contests like LeetCode, AtCoder, and Codeforces. We select this dataset for the **Coding** category.

**Data Filtering and selection.** Each of the datasets mentioned above provides a ground truth answer and an algorithm to evaluate the correctness of model outputs. For instance, MMLU-Pro verifies solutions based on regex string matching. During our pipeline execution, we found that some responses were marked incorrect due to minor formatting issues, even though their solutions were correct. Constructing pairs with these responses is problematic, as the "incorrect" response may simply fail the automated check due to a slight format mismatch. This gives the judge an unintended shortcut, reducing the quality of the dataset.

To address this, we used an additional LLM (GPT-4o-mini) to verify the correctness of solutions. The model was prompted to extract the solution from the response and determine whether it was

correct, regardless of format. We filtered out responses where the LLM and the automated solution checker disagreed. Upon manual inspection of these disagreements, we confirmed they were indeed caused by format errors. An example case where the two methods disagree can be found in Appendix A.5. We perform some additional randomized filtering on MMLU-Pro and LiveBench to better balance the size of each subset (see Appendix A.3 for details).

After applying our pipeline with GPT-4o as the underlying model and incorporating the additional filtering, our dataset consists of a total of 350 questions: 154 in **Knowledge**, 98 in **Reasoning**, 56 in **Mathematics**, and 42 in **Coding**.

## 4 EVALUATION

LLM-based judges are known to exhibit positional bias (Zheng et al., 2024; Wang et al., 2023a), where the order in which the response pairs are presented can influence their decision. Evaluating the judge on a single order of responses introduces this bias into evaluation. To mitigate this, we evaluate the LLM-based judge twice, swapping the order of the response pairs in the second trial.

Since our response pairs contain an objectively correct and incorrect response, the only valid decisions are $A > B$ and $A < B$. However, in practice, some judges support a tie option: $A = B$. To address this discrepancy, we aggregate the results from both trials as follows: if both trials yield $A > B$ or one trial gives $A > B$ and the other $A = B$, we consider the aggregate decision to be $A > B$. Inconsistent decisions (e.g., $A > B$ in one trial, $A < B$ in the other) or ties in both trials are deemed incorrect, as they indicate the judge is either guessing or unable to reliably distinguish between responses. This method enables a more accurate measurement of the judges' ability.

### 4.1 EVALUATING LLM-BASED JUDGES ON JUDGEBENCH.

In this subsection, we describe three major experiments we conduct on JudgeBench. First, we assess LLM-based judges from prior literature, which can be categorized as prompted, fine-tuned, and multi-agent judges. Second, we use JudgeBench as a proxy to evaluate the underlying LLM's performance by fixing the prompt and varying the models. Lastly, we apply JudgeBench to evaluate reward models. In Section 4.2, we provide a detailed analysis of these results.

**Evaluating LLM-based judges across categories.** We evaluate the following three categories of LLM-based judges on JudgeBench. Additional details about these judges can be found in Appendix A.1.

- **Prompted Judges.** For prompted judges, we include the **Vanilla** judge, adapted from AlpacaFarm (Dubois et al., 2024), which directly prompts the LLM to indicate its preferred response without requiring an explanation. We also consider the **Arena-Hard Judge** (Li et al., 2024), which prompts the LLM to first generate its own reference answer, and then analyze both responses before delivering a final verdict. We also include Google's **VertexAI Evaluation** service (Cloud, 2024) in this category.

- **Fine-tuned Judges.** For fine-tuned judges, we evaluate **PandaLM** (Wang et al., 2023c) (fine-tuned on LLaMA-7B (Touvron et al., 2023a)), **Prometheus2** (Kim et al., 2024b) (fine-tuned on Mistral-7B/Mixtral-8x7B), **JudgeLM** (Zhu et al., 2023b) (fine-tuned on Vicuna-7B/13B/33B (Chiang et al., 2023)), **AutoJ** (Li et al., 2023a) (fine-tuned on LLaMA-2-13B-chat (Touvron et al., 2023b)), and **Skywork**'s judges (Shiwen et al., 2024) fine-tuned on Llama-3.1-8B/70B (Dubey et al., 2024). These models are fine-tuned using either crowd-sourced preference datasets or on distilled GPT-4 judgments.

- **Multi-Agents Judges.** For multi-agent judges, we evaluate **ChatEval** (Chan et al., 2023), which leverages multiple LLMs in a debate to produce the final judgment.

**Evaluating JudgeBench on different models.** JudgeBench can also be used as a benchmark to evaluate the underlying model's capability. To evaluate the ability of these models, we freeze the Arena-Hard Judge's prompt and change the underlying model to see how the performance of different models varies. We select the latest models from five model providers: OpenAI, Anthropic,

|  | Knowledge | Reasoning | Math | Coding | Overall |
|---|---|---|---|---|---|
| **Prompted Judges** | | | | | |
| Vanilla (GPT-4o) | 44.16 | 47.96 | 66.07 | **61.90** | 50.86 |
| Arena-Hard Judge (GPT-4o) | 50.65 | 54.08 | **75.00** | 59.52 | 56.57 |
| VertexAI Evaluation (Gemini-1.5-pro) | 45.45 | 44.90 | 53.57 | 28.57 | 44.57 |
| **Fine-tuned Judges** | | | | | |
| PandaLM | 9.09 | 21.43 | 7.14 | 16.67 | 13.14 |
| Prometheus2-7b | 38.31 | 25.51 | 35.71 | 42.86 | 34.86 |
| Prometheus2-8x7b | 41.56 | 39.80 | 50.00 | 23.81 | 40.29 |
| Prometheus2-bgb-8x7b | 45.45 | 30.61 | 46.43 | 28.57 | 39.43 |
| JudgeLM-7B | 23.38 | 29.59 | 32.14 | 11.90 | 25.14 |
| JudgeLM-13B | 26.62 | 29.59 | 28.57 | 19.05 | 26.86 |
| JudgeLM-33B | 32.47 | 48.98 | 33.93 | 19.05 | 35.71 |
| AutoJ | 40.26 | 29.59 | 44.64 | 28.57 | 36.57 |
| Skywork-LLaMA-3.1B-8B | 51.30 | 54.08 | 73.21 | 33.33 | 53.43 |
| Skywork-LLaMA-3.1B-70B | **55.84** | **55.10** | 73.21 | 47.62 | **57.43** |
| **Multi-Agent Judges** | | | | | |
| ChatEval | 32.47 | 31.63 | 44.64 | 30.95 | 34.00 |

Table 1: Evaluating LLM-based judges on JudgeBench.

Meta, Google, and DeepSeek. Some of these models we host ourselves and for others, we rely on either official or third-party APIs. We contain the details of our model sources in Appendix A.1.

| Model | Knowledge | Reasoning | Math | Coding | Overall |
|---|---|---|---|---|---|
| GPT-4o | 50.65 | 54.08 | 75.00 | 59.52 | 56.57 |
| GPT-4o-mini | 48.05 | 43.88 | 69.64 | 45.24 | 50.00 |
| o1-preview | 66.23 | 79.59 | 85.71 | 85.71 | 75.43 |
| o1-mini | 58.44 | 62.24 | 82.14 | 78.57 | 65.71 |
| o3-mini (high) | **67.53** | **89.80** | **87.50** | **100.0** | **80.86** |
| o3-mini (medium) | 62.34 | 86.73 | 85.71 | 92.86 | 76.57 |
| o3-mini (low) | 62.99 | 69.39 | 83.93 | 83.33 | 70.57 |
| Claude-3.5-Sonnet | 62.34 | 66.33 | 66.07 | 64.29 | 64.29 |
| Claude-3-Haiku | 35.06 | 34.69 | 33.93 | 21.43 | 33.14 |
| Llama-3.1-405B-Instruct | 55.84 | 54.08 | 69.64 | 50.00 | 56.86 |
| Llama-3.1-70B-Instruct | 51.30 | 48.98 | 60.71 | 52.38 | 52.29 |
| Llama-3.1-8B-Instruct | 38.31 | 45.92 | 44.64 | 33.33 | 40.86 |
| Gemini-1.5-pro | 49.35 | 42.86 | 64.29 | 26.19 | 47.14 |
| Gemini-1.5-flash | 42.86 | 36.73 | 50.00 | 21.43 | 39.71 |
| Deepseek-R1 | 59.09 | 82.65 | 80.36 | 92.86 | 73.14 |

Table 2: Evaluating the Arena-Hard Judge on JudgeBench, with different underlying models.

**Evaluating JudgeBench on reward models.** While our primary focus is on evaluating LLM-based judges, JudgeBench can also be used to assess reward models, which are trained on preference data to evaluate model outputs. Unlike pairwise LLM-based judges, reward models independently assign scores to each response, and the higher-scoring response is deemed the preferred one.

In our experiments, we evaluated several top-performing reward models from the RewardBench leaderboard (Lambert et al., 2024), including Skywork Reward's model (Liu & Zeng, 2024), InternLM's reward models (Cai et al., 2024), and the GRM-Gemma-2B (Yang et al., 2024) reward model fine-tuned on Google's Gemma model (Team et al., 2024). Results are presented in Table 3

| Reward Model | Knowledge | Reasoning | Math | Coding | Overall |
|---|---|---|---|---|---|
| Skywork-Reward-Gemma-2-27B | 59.74 | 66.33 | 83.93 | 50.00 | 64.29 |
| Skywork-Reward-Llama-3.1-8B | 59.09 | 64.29 | 76.79 | 50.00 | 62.29 |
| InternLM2-20B-Reward | 62.34 | 69.39 | 66.07 | 50.00 | 63.43 |
| InternLM2-7B-Reward | 56.49 | 61.22 | 71.43 | 50.00 | 59.43 |
| GRM-Gemma-2B | 62.99 | 53.06 | 64.29 | 54.76 | 59.43 |

Table 3: Evaluating reward models on JudgeBench.

## 4.2 ANALYSIS AND TAKEAWAYS OF EVALUATION RESULTS ON JUDGEBENCH.

**LLM-based judges' performance falls short under JudgeBench's challenging questions.** The evaluation results from Table 1 and 2 highlight the difficulty of JudgeBench. Even a strong model like GPT-4o struggles, achieving accuracy no better than random guessing when using the vanilla prompt. Although the more advanced Arena-Hard prompt improves performance slightly (from $50\%$ to $56\%$), the overall accuracy remains low.

All of our fine-tuned judges (except Skywork) perform significantly below the random baseline. We explore several reasons behind the relatively poor performance of fine-tuned judges in Appendix A.2. Among all the fine-tuned judges, Skywork's LLM-based judges (Shiwen et al., 2024) perform the best overall, with an overall accuracy of $57.43\%$. When compared to the base Llama-3.1 models in Table 2, fine-tuning shows a clear performance boost, improving accuracy by over $12\%$ for the 8B model and $5\%$ for the 70B model.

From Table 2, we can see that there is a clear gap in the performance between different models, with larger models generally performing better than their smaller counterparts across all providers. Among all models, OpenAI's o3-mini performs the best overall, achieving $80.86\%$, $76.57\%$ and $70.57\%$ accuracy at high, medium, and low reasoning levels respectively. These reasoning-enhanced models differ from standard models by taking extra time to "think" before generating a response (OpenAI, 2024). The superior results indicate that scaling test-time compute is a promising path to improve the reasoning ability of the judges. Beyond these models, Claude-3.5-Sonnet ranks highest among general-purpose models with an accuracy of $64.29\%$. Despite these results being well above the random baseline, all models still have considerable room for improvement.

**Reward models' performance is on par with much more powerful LLMs.** When comparing the performance of reward models to LLM-based judges (Table 1), we find that fine-tuned reward models generally outperform LLM-based judges. For example, Skyworks's Gemma-2-27B reward model achieves accuracy comparable to Claude-3.5-Sonnet, one of the most advanced LLMs currently available, and Skywork's Llama-3.1-8B reward model surpasses the performance of the base model by a huge margin (62.29% vs 40.86%). The above observation indicates that training a specialized verifier from a weak model to judge a stronger model is possible.

Our results also show that reward models exhibit a smaller performance gap on JudgeBench, with overall accuracies ranging from approximately $59\%$ to $64\%$. This consistency is likely due to these models being trained or fine-tuned on similar open preference datasets, such as HelpSteer2 (Wang et al., 2024b) and Skywork-Preference-80 (Skywork, 2023). While model size does influence performance, as seen in the higher accuracy of InternLM-20B compared to InternLM-7B and Skywork's Gemma-2-27B outperforming LLaMA-3.1-8B, the improvements are modest. This suggests that the quality of the training datasets plays a more critical role in shaping the preferences of reward models than model size alone.

**Advancing the reasoning ability of LLM-based judges is the next frontier.** Our evaluation on JudgeBench highlights the limitations of current LLM-based judges in distinguishing between challenging response pairs. As AI systems become more sophisticated, LLM-based judges risk becoming a bottleneck to further scaling. For example, Brown et al. (2024) demonstrated that repeated sampling can improve a model's coverage (the percentage of problems solved) when an oracle-level verifier is available. However, when the verifier is not strong enough, it becomes the limiting factor in this process. Thus, enhancing the reasoning capabilities of LLM-based judges is essential for ad-

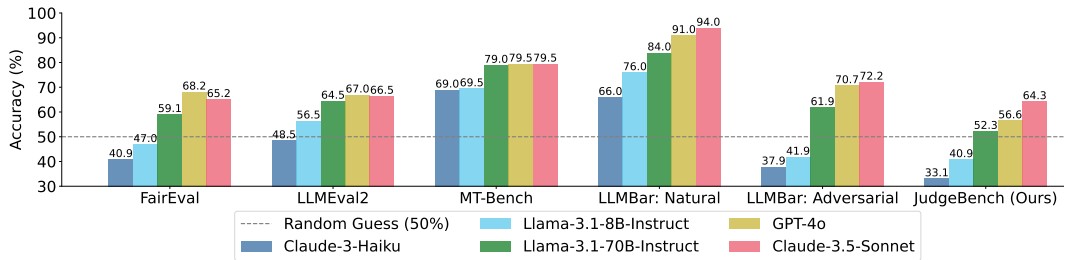

Figure 3: Comparison of JudgeBench against prior benchmarks for LLM-based judges.

vancing the overall performance of AI systems–an area that remains largely underexplored. While we leave improving the LLM-based judges as future work, JudgeBench provides a robust test bed for evaluating future reasoning-enhanced judges.

## 4.3 COMPARING JUDGEBENCH TO OTHER EXISTING BENCHMARKS.

**Comparison with MTBench, LLMEval, FairEval, and LLMBar**  To compare with existing benchmarks for LLM-based judges, we evaluate five models (GPT-4o, Claude-3.5-Sonnet, Llama-3.1-70B-Instruct, Llama-3.1-8B-Instruct, and Claude-3.5-Haiku) on existing datasets using the Arena-Hard prompt. For fairness, we apply the same evaluation procedure described in Section 4, where the judge is run twice on each pair, and its final decision is based on the aggregated judgments.

Our results show that JudgeBench is the most challenging dataset for evaluating LLM-based judges. The strongest model on JudgeBench achieves only 64% accuracy, the lowest among all five datasets. Additionally, JudgeBench demonstrates strong separability, with a 31% performance gap between the best-performing model (Claude-3.5-Sonnet) and the weakest (Claude-3.5-Haiku). This gap is comparable to LLMBar: Adversarial, which has a 33% gap, indicating that JudgeBench is a strong benchmark for evaluating LLM-based judges.

**Comparison with RewardBench.**  RewardBench (Lambert et al., 2024) is a general benchmark for evaluating reward models, with a subcategory dedicated to reasoning in reward models. It includes PRM Math (Lightman et al., 2023) and HumanEvalPack (Muennighoff et al., 2023) as benchmarks for reasoning tasks. These datasets are similar to the Math and Coding categories in JudgeBench, with PRM Math evaluating correct versus incorrect math proofs and HumanEvalPack comparing correct versus buggy code. However, these benchmarks are very saturated, with the strongest model achieving up to 97% accuracy on these datasets. This saturation is likely due to data contamination since datasets such as PRM-800k are widely used in training reward models nowadays. In contrast, JudgeBench is far more challenging, with top reward models achieving only 64% accuracy. Thus, JudgeBench offers a valuable complement to RewardBench for evaluating reward models on difficult tasks requiring reasoning.

## 4.4 ABLATION STUDIES

**Is verifying a problem's solution easier than solving the problem itself?**  Intuitively, verification should be simpler, as the model is provided with candidate solutions and only needs to identify the correct one, a task that would yield 50% accuracy through random guessing alone. To explore this, we conducted an ablation study in which we prompted models to directly solve the problem and compared their accuracy to that of the judge. Our results, presented in Table 4, show that for a fixed model, the judge's accuracy closely mirrors that of the solver. While GPT-4o's and Gemini-1.5-Pro's judges slightly outperform their corresponding solvers, Claude-3.5-Sonnet's and Llama-3.1-405B-Instruct's judges lag behind their respective solvers.

Although the overall accuracy between the solver and judge is close, we observe a notable discrepancy in the Coding category, where the solver consistently outperforms the judge across all models. Conversely, in the Math category, judges significantly outperform solvers. This suggests that coding problems are more difficult to evaluate, while logical errors in math problems are generally easier

| Setup | Knowledge | Reasoning | Math | Coding | Overall |
|---|---|---|---|---|---|
| GPT-4o Solver | 48.70 | 53.06 | 58.93 | 73.81 | 54.57 |
| GPT-4o Judge | 50.65 | 54.08 | 75.00 | 59.52 | 56.57 |
| Claude-3.5-Sonnet Solver | 61.04 | 62.24 | 60.71 | 88.10 | 64.57 |
| Claude-3.5-Sonnet Judge | 62.34 | 66.33 | 66.07 | 64.29 | 64.29 |
| Llama-3.1-405B-Instruct Solver | 48.05 | 67.86 | 63.27 | 66.67 | 57.71 |
| Llama-3.1-405B-Instruct Judge | 55.84 | 54.08 | 69.64 | 50.00 | 56.86 |
| Gemini-1.5-pro Solver | 33.12 | 42.86 | 37.50 | 64.29 | 40.29 |
| Gemini-1.5-pro Judge | 49.35 | 42.86 | 64.29 | 26.19 | 47.14 |

Table 4: Evaluating the LLM's ability to solve the problems.

to identify. Overall, this ablation study indicates that the ability of the judge to verify the solution pairs is highly correlated with its ability to solve the problem itself.

**Investigating bias of the pipeline.**   In JudgeBench's pipeline, we use GPT-4o to generate all response pairs. This introduces a bias against GPT-4o judges, as the generated pairs are inherently challenging for GPT-4o itself to distinguish. To empirically investigate this bias, we conduct an ablation study using Claude-3.5-Sonnet instead to generate response pairs. This results in 270 pairs, 154 pairs for Knowledge, 51 pairs for Reasoning, 34 pairs for Math, and 31 pairs for Coding.

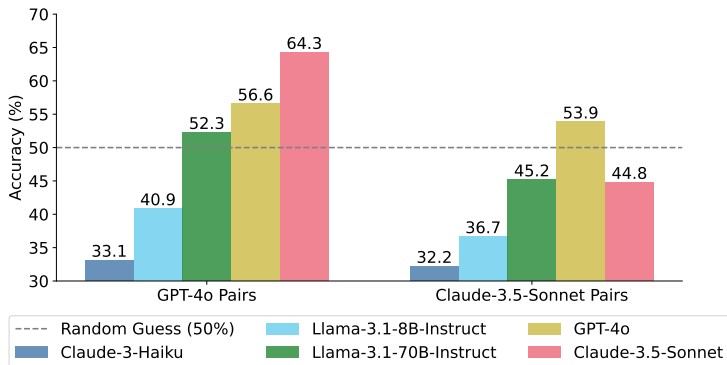

Figure 4: Comparing JudgeBench's evaluation results on GPT-4o versus Claude-3.5-Sonnet generated pairs.

Figure 4 presents a side-by-side comparison of several models' performance on GPT-4o pairs versus Claude-3.5-Sonnet pairs. The results confirm our hypothesis: Claude-3.5-Sonnet, which achieves 64.3% accuracy on GPT-4o pairs, drops to 44.8% accuracy when tasked with judging its own generated pairs. Similarly, GPT-4o tops the Claude-3.5-Sonnet pairs with 53.9% accuracy. However, this number is still slightly lower than the 56.6% accuracy on its own pairs. This suggests that Claude-3.5-Sonnet is a stronger reasoning model, producing more difficult-to-distinguish response pairs in general. Beyond the models used to generate the response pairs, other models exhibit similar performance gaps, indicating that the response pairs remain consistent in evaluating model capabilities, regardless of the model used to generate the pairs.

## CONCLUSION

In this work, we introduce a novel hierarchical evaluation framework to objectively evaluate LLM-based judges. Based on this framework, we propose JudgeBench, a benchmark for evaluating LLM-based judges' ability to distinguish factually and logically correct outputs. Our work addresses the pressing need to evaluate LLM-based judges' reasoning ability, which is of increasing importance given the rapid advancement of AI intelligence today. We hope that our framework and benchmark can offer insights into future dataset design and foster further research into this space.

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

# A APPENDIX

## A.1 DETAILS OF THE JUDGES

We closely followed the official implementation of each judge, only making modifications where necessary. One broad change we made across all judges is the use of greedy decoding (temperature=0) to ensure reproducibility. Any additional judge-specific modifications are detailed below. We sourced proprietary LLMs through their official APIs. All open-weight LLMs (including reward models) were served locally in half-precision, except for Llama-3.1-405B-Instruct for which we utilized the TogetherAPI. The prompts for each judge are provided for reference in Appendix A.6.

### A.1.1 PROMPTED JUDGES

**Vanilla** (Dubois et al., 2024): This is a basic judge prompted to output a label indicating which response it believes to be better, with no explanation required. This judge has no tie option. Each judgment must be no more than 1024 tokens, however, in practice, they contained significantly fewer tokens.

**Arena-Hard Judge** (Li et al., 2024): This judge is used in LMSYS's Arena-Hard Leaderboard[1]. It's prompted to provide its own response to the question to use as a reference before evaluating the pair of candidate responses. This judge must decide between 5 options: A>>B, A>B, A=B, B>A, or B>>A. We did not distinguish between the first two cases, nor did we distinguish between the last two cases. Following the original implementation, each judgment must be less than 4096 tokens; however, if we were unable to extract the verdict using regex (e.g., if the judgment was incomplete after 4096 tokens), the judge was given one more opportunity to continue its judgment (up to 4096 additional tokens) and output a valid verdict. One special case worthy of note is the Arena-Hard results with o1-mini and o1-preview may not have respected these token constraints nor the zero temperature.

**Google Vertex Judge** (Cloud, 2024): Google Cloud offers a generative AI evaluation service in Vertex AI powered by Gemini-Pro-001. It supports both single and pairwise evaluation, though we only evaluated in the pairwise configuration using their predefined question-answering quality metric. The service is proprietary and offers little to no ability to set generation parameters (e.g., temperature).

### A.1.2 FINE-TUNED JUDGES

**PandaLM** (Wang et al., 2023c): PandaLM is a family of LLM-based judges based on LLaMA-7B and LLaMA-70B (Touvron et al., 2023a), and fine-tuned on crowdsourced human preference data collected by the authors. As of the time of publication, only the 7B variant of PandaLM has been made publicly accessible, so we do not include results on the 70B variant. PandaLM supports a tie option. We followed PandaLM's official implementation closely, including beam searching over 4 beams. We made one crucial change to PandaLM's official inference pipeline: we truncated both candidate responses (from the left) to fit the request in the limited context window of 2048 tokens. Left truncation was used as many of the responses in JudgeBench output their final decision at the end; experimentally we found that left truncation performs better than right truncation. Although PandaLM generates its decision before its explanation, we generated up to 150 tokens to give the beams time to "mature."

**Prometheus 2** (Kim et al., 2024b): Prometheus 2 is a family judges fine-tuned from Mistral 7B (Jiang et al., 2023) and Mixtral 8x7B (Jiang et al., 2024). These models are fine-tuned separately on both Feedback Collections (Kim et al., 2023), a direct-assessment synthetic dataset generated by GPT-4 (OpenAI et al., 2024), and Preference Collections (Kim et al., 2024b), an augmented version of Feedback Collections for pairwise evaluation, with the resulting weights merged. Additionally, the authors subsequently released a second Mixtral variant further trained on the BiGGen Bench (Kim et al., 2024a). We evaluated all 3 Prometheus 2 models on JudgeBench. Prometheus 2

---

[1]https://huggingface.co/spaces/lmsys/arena-hard-browser

judges support fine-grained evaluation criteria, and we used their official "factual validity" criteria since it best aligns with the motivation of JudgeBench. Prometheus 2 does not support ties.

**JudgeLM** (Zhu et al., 2023b): JudgeLM is a family of judges fine-tuned from Vicuna (Chiang et al., 2023) using a dataset collected from existing instruction-tuning datasets that have been augmented with candidate responses and GPT-4 (OpenAI et al., 2024) judgments. JudgeLM assigns integer scores to each candidate response, meaning ties are possible albeit unlikely. We made one crucial change to JudgeLM's official inference pipeline: we truncated both candidate responses (from the left) to fit the request in the limited context window of 2048 tokens. Left truncation was used as many of the responses in JudgeBench output their final decision at the end; experimentally we found that left truncation performs better than right truncation. Since JudgeLM generates scores before an explanation, we limited the number of generated tokens to just 16 to reduce the amount that each candidate response is truncated.

**Auto-J** (Li et al., 2023a): Auto-J is a generative judge fine-tuned from Llama-2-13b-chat (Touvron et al., 2023b) on publicly available preference datasets augmented with GPT-4 (OpenAI et al., 2024) judgments. Auto-J supports ties. Following Auto-J's official implementation, we generated judgments up to 1024 tokens in length at a temperature of 0.

**Skywork Critics** (Shiwen et al., 2024): Skywork released a series of 8B and 70B generative judges built on Llama-3.1-8B-Instruct and Llama-3.1-70B-Instruct (Dubey et al., 2024) respectively. These models are fine-tuned on a combination of proprietary and open-source critic datasets. Following their official implementation, we generated judgments up to 2048 tokens at a temperature of 0.

### A.1.3 MULTI-AGENT JUDGES

**ChatEval** (Chan et al., 2023): ChatEval is a multi-agent judge that assigns roles to each agent. Our implementation used two agents (one to act at the general public, while the other acts as a critic) powered by GPT-4o as the model. The agents discuss sequentially (in a round-robin fashion) for at most 4 turns. After the discussion, each agent independently assigns a score (between 1 and 10) for each candidate response. We averaged the scores across both agents to determine the final decision.

## A.2 ADDITIONAL ANALYSIS OF FINE-TUNED JUDGES

Many of the fine-tuned judges we evaluate score below the random guessing baseline of 50%. In this section, we highlight three reasons this is the case: (1) truncated responses, (2) ties and invalid decisions, and (3) inconsistent judgments. Moreover, we compare a few fine-tuned judges with pure-prompting judges using the same base model to understand the impact of fine-tuning for judging.

**Truncated responses** Two of our judges (PandaLM and JudgeLM) have limited context windows supporting just 2048 tokens. Combined, our candidate responses, however, often exceeded this limit. As such, we dynamically truncated (from the left) both candidate responses (leaving all other parts of the prompt template unchanged) to ensure the requests fit within the context limit. First, since many candidate responses do not output their final answer until the end of their responses, truncating from the left ensures the final answers are included in the truncated response provided to the judge. Second, early experimentation with PandaLM and JudgeLM revealed that truncating from the left resulted in better performance than truncating from the right.

**Ties and invalid decisions** Closely inspecting the generations of our fine-tuned judges reveals several weaknesses. For example, out of 700 judgments (2 games across 350 examples), PandaLM selected the tie option 479 times. Another Judge, Prometheus2-bgb-8x7b, made an invalid judgment from which we cannot extract a decision on 215 of 700 judgments. Some of these invalid judgments included "10/10", "Neither A nor B", and "3". Table 5 provides the number of occurrences (out of 700 games) each of the fine-tuned judges selects $A > B$, $A < B$, $A = B$, or the judgment is otherwise invalid.

| Judge | $A > B$ | $A < B$ | $A = B$ | Invalid |
|---|---|---|---|---|
| PandaLM-7B | 45 | 114 | 479 | 62 |
| Prometheus2-7b | 395 | 232 | 0 | 73 |
| Prometheus2-8x7b | 331 | 328 | 0 | 41 |
| Prometheus2-bgb-8x7b | 239 | 215 | 0 | 246 |
| JudgeLM-7B | 399 | 229 | 72 | 0 |
| JudgeLM-13B | 355 | 312 | 33 | 0 |
| JudgeLM-33B | 344 | 264 | 92 | 0 |
| AutoJ | 289 | 378 | 33 | 0 |
| Skywork-LLaMA-3.1B-8B | 346 | 354 | 0 | 0 |
| Skywork-LLaMA-3.1B-70B | 390 | 310 | 0 | 0 |

Table 5: The prevalence of judgment types for each fine-tuned judge.

| Judge | Inconsistent |
|---|---|
| PandaLM-7B | 29.14% |
| Prometheus2-7b | 52.29% |
| Prometheus2-8x7b | 40.00% |
| Prometheus2-bgb-8x7b | 43.71% |
| JudgeLM-7B | 59.71% |
| JudgeLM-13B | 54.57% |
| JudgeLM-33B | 38.00% |
| AutoJ | 43.71% |
| Skywork-Llama-3.1B-8B | 18.86% |
| Skywork-Llama-3.1B-70B | 18.29% |

Table 6: Rate of inconsistency between trials for each fine-tuned judge.

**Inconsistent judgments**    Many fine-tuned judges struggled to generate consistent results between games. For example, JudgeLM-7B and JudgeLM-13B were inconsistent on 59.71% and 54.57% of pairs respectively. Likewise, Prometheus2-7b was inconsistent on 52.29% of pairs. Table 6 shares the rate of inconsistency in judgments between games.

**Case Study: Do fine-tuned judges outperform prompted judges?**    The fine-tuned Skywork-Llama-3.1B-8B and Skywork-LLaMA-3.1B-70B outperformed the corresponding arena-hard judges with the same base model (Llama-3.1B-8B-Instruct and Llama-3.1B-70B-Instruct) by 12.57 and 5.14 respectively. Does this hold for other judges? In an attempt to answer this question, we evaluated Mistral-7B-v0.1-Instruct, the base model behind Prometheus2-7b, using the Vanilla and Arena-Hard prompts and present the results in Table 7. We found that Prometheus2-7b significantly outperformed its base model (Mistral-7B-v0.1-Instruct) with both the vanilla prompt and Arena-Hard prompt. It's worth noting that with the vanilla prompt, Mistral-7B-v0.1-Instruct selected response A in 624/700 games, and with the arena-hard prompt, Mistral-7B-v0.1-Instruct selected the tie option in 618/700 games. As such, it does appear that fine-tuned judges tend to outperform prompted judges when using the same base model.

| Judge | Score |
|---|---|
| Prometheus2-7b | 38.31 |
| Vanilla (Mistral-7B-v0.1-Instruct) | 7.43 |
| Arena-Hard (Mistral-7B-v0.1-Instruct) | 6.57 |

Table 7: Comparison of Prometheus2-7b against prompted judges with the same base model.

A.3 DATASET FILTERING

We performed some additional filtering on MMLU-Pro and LiveBench. For MMLU-Pro, we randomly selected 100 questions from each of the 14 disciplines before generating responses and constructing pairs. Of these 1400 questions, only 347 and 233 contained both correct and incorrect responses generated by GPT-4o and Claude-3.5-Sonnet, respectively. In both cases, we randomly selected 11 pairs from each discipline, for a total of 154 knowledge pairs. Similarly, we randomly sampled 100 questions from the math and reasoning subsets of Livebench. Note that we exclude the "olympiad" subset of LiveBench-Math. We derived 98 reasoning pairs and 46 math pairs from GPT-4o responses, but just 51 reasoning pairs and 34 math pairs from Claude-3.5-Sonnet responses. We did no pre-filtering or post-filtering for LiveCodeBench, deriving 42 pairs from GPT-4o responses and 31 pairs from Claude-3.5-Sonnet responses.

In all, the GPT-4o split of JudgeBench contains 350 instances, which is on par with similar benchmarks. For instance, FairEval (Wang et al., 2023a) contains 80 unique questions, LLMEval-2 (Zhang et al., 2023) contains 480, MT-Bench (Zheng et al., 2024) contains 80, and LLMBar (Zeng et al., 2023) contains 419. RewardBench (Lambert et al., 2024) is larger, but it's an aggregation of existing benchmarks, including MT-Bench and LLMBar. In order to test if this size is sufficient, we augmented our "knowledge" subset, increasing the number of response pairs from 154 to 770. We evaluated several LLMs using the Arena-Hard prompt, and observed that the relative rankings among these judges were the same between our original set and the augmented set, despite small variations in the scores themselves (see Table 8).

| Model | Original Set | Augmented Set |
|---|---|---|
| gpt-4o | 50.65 *(3rd)* | 46.49 *(3rd)* |
| gpt-4o-mini | 48.05 *(4th)* | 44.03 *(4th)* |
| claude-3.5-sonnet | 62.34 *(1st)* | 63.25 *(1st)* |
| claude-3-haiku | 35.06 *(6th)* | 39.35 *(6th)* |
| llama-3.1-70b-instruct | 51.30 *(2nd)* | 52.60 *(2nd)* |
| llama-3.1-8b-instruct | 38.31 *(5th)* | 40.00 *(5th)* |

Table 8: Performance on original and augmented "knowledge" sets.

A.4 ANALYSIS OF LENGTH BIAS ON JUDGE BENCH

Prior works has documented that LLM-based judges exhibit length bias, tending to prefer longer responses over shorter ones (Hu et al., 2024; Wei et al., 2024). Because each of our pairs contains two responses sampled from the same model, rather than from two different models, the responses tend to be of similar length. On average, across all 350 instances of JudgeBench, correct and incorrect responses contain 562.29 and 561.16 tokens, respectively, using the GPT-4o tokenizer. This negligible difference demonstrates that the construction of JudgeBench effectively mitigates length bias, allowing LLM-based judges to be evaluated without this confounding factor.

A.5 EVALUATING RESPONSES FOR CORRECTNESS

For LiveBench and MMLU-Pro, we checked the correctness of generated responses using two methods. First, we parsed the final answer from the responses using regex and checked against the ground truth answers. For LiveBench, we closely followed their official post-processing methodology to extract the final answers. For MMLU-Pro, we adapted the multiple-choice questions to the same format used by LiveBench and included in the question instructions to output its final letter choice 5 times in a row (e.g., "My final answer is AAAAA"). Using regex, we extracted the last capital letter A-J that was repeated 5 times and treated this as the final answer. Second, we queried GPT-4o-mini with the question, response, and ground truth (using the prompt provided below) and had it decide whether or not the response was correct. We excluded any responses where these two methods disagree.

---

**Prompt template for checking correctness with GPT-4o-mini.**

**System Prompt:**
I will provide you with a multiple-choice question, a response from an LLM, and the correct option. Output a valid JSON object containing a single key-value pair, where the key is "is_correct" and corresponding value is a boolean indicating whether or not the LLM-generated selects the correct option.

**User Prompt:**
<|Question|>
{question}

<|LLM Response|>
{response}

<|Correct Answer|>
{ground_truth}

---

The advantage of this combined approach is to eliminate any questionable responses, such as in the case where the model selects the correct answer but does not properly format its answer in accordance with the instructions. In general, we found high rates of agreement between these methods (e.g., 97.7% and 99.5% agreement across 7000 responses to the 14000 MMLU-Pro questions for GPT-4o and Claude-3.5-Sonnet, respectively). Below, we highlight an example case of disagreement, where the model correctly identified the correct answer, but formatted it incorrectly as `IIIIi` instead of `IIIII`, which resulted in no regex match but GPT-4o-mini gave it the benefit of the doubt and marked it correct.

---

**Example of disagreement between solution checkers.**

**Question:**
Which muscle is the most active during a right lateral excursion of the mandible?
(A) Left masseter muscle
(B) Right buccinator muscle
(C) Right temporalis muscle
(D) Right masseter muscle
(E) Right lateral pterygoid muscle
(F) Left medial pterygoid muscle
(G) Left temporalis muscle
(H) Left buccinator muscle
(I) Left lateral pterygoid muscle
(J) Right medial pterygoid muscle
If you cannot determine the correct multiple-choice answer, take your best guess. Once you have your answer, please duplicate that letter five times in a single string. For example, if the answer is K, then write KKKKK.
Let's think step by step.

**Response:**
To determine which muscle is most active during a right lateral excursion of the mandible, we need to understand the movements and the muscles involved. A right lateral excursion means the mandible is moving to the right.

1. **Muscles Involved in Mandibular Movements**:
- The **lateral pterygoid muscles** are the main muscles responsible for the lateral movements of the mandible. Each side's lateral pterygoid has a different function during this movement.
- The **right lateral pterygoid muscle** would pull the condyle of the mandible forward on the right side, resulting in leftward movement.

---

- The **left lateral pterygoid muscle** would cause the condyle on the left to move forward, assisting the mandible in moving to the right.

2. **Supporting Muscles**:
- The **masseter, temporalis, medial pterygoid**, and **buccinator** play roles primarily in elevation, retraction, or mastication rather than direct lateral movements.

3. **Logical Deduction**:
- When the mandible moves to the right, the **left lateral pterygoid muscle** is more engaged. This is because it's the left lateral pterygoid that pulls the left side of the mandible forward and to the right.

Given this, the most active muscle during a right lateral excursion of the mandible is: I. Left lateral pterygoid muscle. Therefore, the correct answer is: IIIIi

**Ground Truth:**
(I) Left lateral pterygoid muscle

## A.6 PROMPTS

Below, we share the prompts used for each judge we evaluated. These prompts come directly from each judges' official implementation, but are provided for reference.

---

### Vanilla Prompt

**User Prompt:**
You are a helpful assistant in evaluating the quality of the outputs for a given instruction. Your goal is to select the best output for the given instruction.
Select the Output (a) or Output (b) that is better for the given instruction. The two outputs are generated by two different AI chatbots respectively.
Do NOT provide any explanation for your choice.
Do NOT say both / neither are good.
You should answer using ONLY "Output (a)" or "Output (b)". Do NOT output any other words.
# Instruction:
{question}
# Output (a):
{response_a}
# Output (b):
{response_b}
# Which is better, Output (a) or Output (b)? Your response should be either "Output (a)" or "Output (b)":

---

### Arena-Hard Prompt

**System Prompt:**
Please act as an impartial judge and evaluate the quality of the responses provided by two AI assistants to the user prompt displayed below. You will be given assistant A's answer and assistant B's answer. Your job is to evaluate which assistant's answer is better.

Begin your evaluation by generating your own answer to the prompt. You must provide your answers before judging any answers.

When evaluating the assistants' answers, compare both assistants' answers with your answer. You must identify and correct any mistakes or inaccurate information.

Then consider if the assistant's answers are helpful, relevant, and concise. Helpful means the answer correctly responds to the prompt or follows the instructions. Note when user prompt has any ambiguity or more than one interpretation, it is more helpful and appropriate to ask for clarifications or more information from the user than providing an answer based on assumptions. Relevant means all parts of the response closely connect or are appropriate to what is being asked. Concise means the response is clear and not verbose or excessive.

Then consider the creativity and novelty of the assistant's answers when needed. Finally, identify any missing important information in the assistants' answers that would be beneficial to include when responding to the user prompt.

After providing your explanation, you must output only one of the following choices as your final verdict with a label:

1. Assistant A is significantly better: [[A>>B]]
2. Assistant A is slightly better: [[A>B]]
3. Tie, relatively the same: [[A=B]]
4. Assistant B is slightly better: [[B>A]]
5. Assistant B is significantly better: [[B>>A]]

Example output: "My final verdict is tie: [[A=B]]".

**User Prompt:**
< | User Prompt | >
{question}

< | The Start of Assistant A's Answer | >
{response_a}
< | The End of Assistant A's Answer | >

< | The Start of Assistant B's Answer | >
{response_b}
< | The End of Assistant B's Answer | >

---

## Google Vertex Prompt

**User Prompt:**
# Instruction
You are an expert evaluator. Your task is to evaluate the quality of the responses generated by two AI models. We will provide you with the user input and a pair of AI-generated responses (Response A and Response B). You should first read the user input carefully for analyzing the task, and then evaluate the quality of the responses based on the Criteria provided in the Evaluation section below.

You will first judge responses individually, following the Rating Rubric and Evaluation Steps. Then you will give step-by-step explanations for your judgment, compare results to declare the winner based on the Rating Rubric and Evaluation Steps.

# Evaluation
## Metric Definition
You will be assessing question answering quality, which measures the overall quality of the answer to the question in the user prompt. Pay special attention to length constraints, such as in X words or in Y sentences. The instruction for performing a question-answering task is provided in the user prompt. The response should not contain information that is not present in the context (if it is provided).

## Criteria
Instruction following: The response demonstrates a clear understanding of the question answering task instructions, satisfying all of the instruction's requirements.
Groundedness: The response contains information included only in the context if the context is present in the user prompt. The response does not reference any outside information.
Completeness: The response completely answers the question with sufficient detail.
Fluent: The response is well-organized and easy to read.

## Rating Rubric
"A": Response A answers the given question as per the criteria better than response B.
"SAME": Response A and B answers the given question equally well as per the criteria.
"B": Response B answers the given question as per the criteria better than response A.

## Evaluation Steps
STEP 1: Analyze Response A based on the question answering quality criteria: Determine how well Response A fulfills the user requirements, is grounded in the context, is complete and fluent, and provides assessment according to the criterion.
STEP 2: Analyze Response B based on the question answering quality criteria: Determine how well Response B fulfills the user requirements, is grounded in the context, is complete and fluent, and provides assessment according to the criterion.
STEP 3: Compare the overall performance of Response A and Response B based on your analyses and assessment.
STEP 4: Output your preference of "A", "SAME" or "B" to the pairwise_choice field according to the Rating Rubric.
STEP 5: Output your assessment reasoning in the explanation field.

# User Inputs and AI-generated Responses
## User Inputs
### Prompt
{question}

# AI-generated Response

### Response A
{response_a}

### Response B
{response_b}

---

**PandaLM Prompt**

**Prompt:**
Below are two responses for a given task. The task is defined by the Instruction. Evaluate the responses and generate a reference answer for the task.

### Instruction:
{question}

### Response 1:
{response_a}

### Response 2:
{response_b}

### Evaluation:

---

**Prometheus 2 Prompt**

**System Prompt:**
You are a fair judge assistant assigned to deliver insightful feedback that compares individual performances, highlighting how each stands relative to others within the same cohort.

**User Prompt:**
###Task Description:
An instruction (might include an Input inside it), a response to evaluate, and a score rubric representing a evaluation criteria are given.
1. Write a detailed feedback that assess the quality of two responses strictly based on the given score rubric, not evaluating in general.
2. After writing a feedback, choose a better response between Response A and Response B. You should refer to the score rubric.
3. The output format should look as follows: "(write a feedback for criteria) [RESULT] (A or B)" 4. Please do not generate any other opening, closing, and explanations.

###Instruction:
{question}

###Response A:
{response_a}

###Response B:
{response_b}

###Score Rubric:
[Are the model's responses factually correct and well-supported by evidence?]

###Feedback:

---

**JudgeLM Prompt**

**Prompt:**
You are a helpful and precise assistant for checking the quality of the answer.
[Question]
{question}

[The Start of Assistant 1's Answer]
{response_a}

[The End of Assistant 1's Answer]

[The Start of Assistant 2's Answer]
{response_b}

[The End of Assistant 2's Answer]

[System]
We would like to request your feedback on the performance of two AI assistants in response to the user question displayed above.
Please rate the helpfulness, relevance, accuracy, level of details of their responses. Each assistant receives an overall score on a scale of 1 to 10, where a higher score indicates better overall performance.
Please first output a single line containing only two values indicating the scores for Assistant 1 and 2, respectively. The two scores are separated by a space. In the subsequent line, please provide a comprehensive explanation of your evaluation, avoiding any potential bias and

ensuring that the order in which the responses were presented does not affect your judgment.

### Response:

---

**Auto-J Prompt**

**User Prompt:**
You are assessing two submitted responses on a given user's query and judging which response is better or they are tied. Here is the data:

[BEGIN DATA]
***
[Query]: {question}
***
[Response 1 ]: {response_a}
***
[Response 2 ]: {response_b}
***
[END DATA]

Here are the instructions to assess and compare the two responses:

1. Pinpoint the key factors to distinguish these two responses.
2. Conclude your comparison by providing a final decision on which response is better, or they are tied. Begin your final decision statement with "So, the final decision is Response 1 / Response 2 / Tie". Ensure that your decision aligns coherently with the comprehensive evaluation and comparison you've provided.

---

**Skywork Prompt**

**User Prompt:**
Please act as an impartial judge and evaluate the quality of the responses provided by two AI assistants to the user question displayed below. You should choose the assistant that follows the user's instructions and answers the user's question better.
Your evaluation should consider factors such as the helpfulness, relevance, accuracy, depth, creativity, and level of detail of their responses. Avoid any position biases and ensure that the order in which the responses were presented does not influence your decision. Do not allow the length of the responses to influence your evaluation. Do not favor certain names of the assistants. Be as objective as possible.
Please directly output your final verdict by strictly following this format: "[[A]]" if assistant A is better, "[[B]]" if assistant B is better.

[User Question]
{question}

[The Start of Assistant A's Answer]
{response_a}
[The End of Assistant A's Answer]

[The Start of Assistant B's Answer]
{response_b}
[The End of Assistant B's Answer]

**ChatEval Prompt**

**General Public Prompt:**
[Question]
{question}

[The Start of Assistant 1's Answer]
{response_a}
[The End of Assistant 1's Answer]
[The Start of Assistant 2's Answer]
{response_b}
[The End of Assistant 2's Answer]
[System]
We would like to request your feedback on the performance of two AI assistants in response to the user question displayed above. Please consider the helpfulness, relevance, accuracy, and level of detail of their responses. There are a few other referees assigned the same task; it's your responsibility to discuss with them and think critically before you make your final judgment. Each assistant receives an overall score on a scale of 1 to 10, where a higher score indicates better overall performance.

You are now General Public, one of the referees in this task. You are interested in the story and looking for updates on the investigation. Please think critically by yourself and note that it's your responsibility to choose one of which is the better first.

Now it's your time to talk, please make your talk short and clear, General Public !

Please first provide a comprehensive explanation of your evaluation, avoiding any potential bias and ensuring that the order in which the responses were presented does not affect your judgment. Then, output two lines indicating the scores for Assistant 1 and 2, respectively.

Remember that you are not required to output the same value as other referees!
Output with the following format strictly:
Evaluation evidence: [your explanation here]
The score of Assistant 1: [score only]
The score of Assistant 2: [score only]

**Critic Prompt:**
[Question]
{question}

[The Start of Assistant 1's Answer]
{response_a}
[The End of Assistant 1's Answer]
[The Start of Assistant 2's Answer]
{response_b}
[The End of Assistant 2's Answer]
[System]
We would like to request your feedback on the performance of two AI assistants in response to the user question displayed above. Please consider the helpfulness, relevance, accuracy, and level of detail of their responses. There are a few other referees assigned the same task; it's your responsibility to discuss with them and think critically before you make your final judgment. Each assistant receives an overall score on a scale of 1 to 10, where a higher score indicates better overall performance.

You are now Critic, one of the referees in this task. You will check fluent writing, clear sentences, and good wording in summary writing. Your job is to question others judgment to make sure their judgment is well-considered and offer an alternative solution if two

responses are at the same level.

Now it's your time to talk, please make your talk short and clear, Critic!

Please first provide a comprehensive explanation of your evaluation, avoiding any potential bias and ensuring that the order in which the responses were presented does not affect your judgment. Then, output two lines indicating the scores for Assistant 1 and 2, respectively.

Remember that you are not required to output the same value as other referees!
Output with the following format strictly:
Evaluation evidence: [your explanation here]
The score of Assistant 1: [score only]
The score of Assistant 2: [score only]

