# OpenReview forum: "JudgeBench: A Benchmark for Evaluating LLM-Based Judges"
_ICLR.cc/2025/Conference — ICLR 2025 Poster_

### Official Review · Reviewer_VLGm · 2024-10-29

**Soundness:** 2
**Presentation:** 3
**Contribution:** 2
**Rating:** 6
**Confidence:** 3

**Summary:**

Currently, many approaches try to employ LLM-Based judges as a cheap and scalable alternative to human judges. However, their limitations remain unknown. While most previous works regard crowdsourced human annotations as ground-truth answers, these evaluations may have mistakes if the questions grow complex in reasoning.
Therefore, the authors present JudgeBench, a benchmark to evaluate LLM-based judges, prioritizing factual and logical correctness. Firstly, the authors propose a pipeline for constructing such a dataset: Given a dataset with ground truth answer labels, they sample k responses from a strong model (e.g., GPT-4o). Then, they filter out questions where all k response are either all correct or all incorrect. They construct response pairs (correct and incorrect) from the remaining questions. Secondly, the pipeline is used to collect a dataset in four distinct categories: knowledge, reasoning, mathematics, and coding. A total of 350 questions are collected.
The resulting dataset is the most challenging for evaluating LLM-based judges. The authors evaluated a number of LLM-based judges on the dataset, ranging from vanilla judges to fine-tuned judges. The best model has a 57.43 performance overall.

**Strengths:**

1. The authors include extensive baselines of critical models and methods. The results lead to intriguing insights on how LLM-based judges fall short in evaluating challenging questions.
2. The motivation is very clear, which is creating a challenging benchmark to assess LLM judge's abilities in evaluating complex and reasoning-heavy problems.

**Weaknesses:**

While the experiments generally show that the resulting benchmark is challenging and has good separability, I believe there are more points that could be further dug into. The limited analysis makes the paper's contribution less significant.

1. The "key insight" mentioned in L229-231 is not verified carefully. The authors state: "if a model struggles to consistently generate correct, coherent responses to a challenging question, it will find it difficult to differentiate between those responses." From Table 4, this doesn't seem to be the case for all setups, especially for reasoning, math, and coding, where we see that stronger models can be weaker judges (Claude-3.5 and Llama-3.1 in reasoning).
2. Also, in Table 1, we see that many LLMs-judges perform less than random-chance. For example, JudgeLM gets a 11.96 accuracy in evaluating coding questions. Does this mean it always predicts the opposite answer? Because that would be another type of bias. There is no failure case analysis or type analysis for these LLM-based judges.
3. In paper [1], the authors notice that reasoning-heavy questions are hard for LLM-based judges to evaluate. Therefore, it is better for them to first construct its own reference answer, and then evaluate the candidate answer. They call this the reference-based judge. Have you considered similar simple prompting strategies to improve performance? How would they perform?
4. In the section on L426, the authors simply state that advancing the reasoning ability of LLM-based judges could help increase performance. This is a rather generic conclusion. It would be nice to have more contributions in producing a better LLM judge that could score higher on JudgeBench, using the insights that the authors have discovered. If not so, this paper stands only on the analysis level. And the analysis is not so deep.
5. The amount of questions collected (350) is small. Could you show the confidence intervals of the resulting estimations?

Reference:
[1] Judging LLM-as-a-Judge with MT-Bench and Chatbot Arena

**Questions:**

1. For your description in L286-298 on position bias: as this is a benchmarking dataset, you also allow room for ambiguity and position bias in LLM-based judges. Wouldn't this also be an aspect to evaluate? I.e., if the judgment results change, the judges should be discredited on their discrepancy. There could be a metric to quantify such position bias.

see weaknesses.

---

> ### Author Response · Authors · 2024-11-18
> **To Reviewer VLGm**
>
> We thank the reviewer for your valuable feedback.
>
> **On judges performing worse than random guessing.** Please see "Response to common comments" for our response to this. Also, we provide a failure case analysis in Appendix A.2 in which we explains why many of the fine-tuned judges performs poorly. Please refer to "On failure cases" above for more details as well.
>
> **On key insight not verified carefully.** Please see "Response to common comments" above for how this insight is verified through Figure 4. Our key insight states that if a model cannot generate consistently correct responses (sampled with a temperature of 1.0) across k trials, then the **same** model will struggle to distinguish between these k responses. Table 4, on the other hand, pertains to a different experimental setting in which we study the performance of several models on solving (greedy decoding, one trial) the set of questions identified via our key insight (i.e., these were only questions which GPT-4o struggled to consistently answer correctly). In this context, the solver's accuracy is not indicative of the difficulty in distinguishing between response pairs. Instead, the key takeaway from Table 4 is that idetifying a correct response in JudgeBench is highly correlated with, and nearly as difficult as, solving the underlying problem itself. This reinforces the challenging nature of our dataset.
>
> **On reference-based judges.** Our study already incorporates reference-based judges in the form of the Arena-hard judge, which is one of the two prompting methods we evaluate: Vanilla judge and Arena-hard judge. The Arena-hard judge, detailed in Appendix A.5 L1080, operates exactly as the reference-based judge mentioned in the feedback. It first generates its own reference solution before making judgments, unlike the Vanilla judge, which directly outputs a judgment. When comparing Arena-hard judge to Vanilla judge (See Table 1), we see a notable accuracy improvement (from 50.86% to 56.57%), showing that reference-based judges do improve performance on reasoning tasks.
>
> **On conclusion from L426 being generic.** We appreciate the feedback regarding the need for more detailed discussions on improving judges' reasoning ability. While we leave improving the LLM-based judges as future work, we expand here on some potential approaches informed by the insights gained from JudgeBench.
>
> * **Agentic judges with tool-use ability.** For complex tasks like math and coding, incorporating external tools such as calculators or code interpreters can facilitate more accurate verification. The challenge lies in designing AI agents that can extract intermediate results from prover responses, apply external tools effectively, and synthesize the tool outputs into final judgments. This approach could significantly enhance reasoning accuracy.
> * **Adversarial training for verifiers.** Our dataset can be viewed as "adversarial", in a sense that we explicitly design it to produce response pairs that are difficult to distinguish. A promising direction involves adapting adversarial machine learning techniques to train robust verifiers. For instance, recent work [1] has explored training provers to generate both valid and misleading proofs, with verifiers trained to distinguish between them. While the focus of this work is on improving the legibility of prover's response and only evaluates on the GSM8k dataset, the idea could be applied to train a stronger verifier on general reasoning problems.
>
> We will integrate these discussion into the section.
>
> [1] https://arxiv.org/pdf/2407.13692
>
> **On evaluating positional bias.** While we do not explicitly measure positional bias, our evaluation method inherently accounts for it. As described, we aggregate the judge's decisions across two independent trials with swapped response orders. If the judge's responses are inconsistent (e.g., A > B in one trial and B > A in the other), the aggregate decision is recorded as A = B, which is deemed incorrect. This discrepency is thus directly discredited in our accuracy metric. Furthermore, in our failure case analysis (Appendix A.2), we measure the ratio of inconsistent responses from judges. This metric can indeed serve as an indicator to quantify positional bias, as you suggest.

---

> > ### Author Response · Authors · 2024-11-23
> >
> > Dear reviewer VLGm,
> >
> > Thank you for your thoughtful review and feedback on our paper. We hope our responses have adequately addressed your concerns.
> >
> > As the author-reviewer communication window is coming to an end, please let us know if you have any additional questions or comments. We would be happy to provide further clarification.
> >
> > We greatly appreciate the time and effort you have dedicated to reviewing our work.
> >
> > Best regards, The JudgeBench Team

---

> > ### Comment · Reviewer_VLGm · 2024-11-25
> > **Reply**
> >
> > Thank you for the reply. I have raised my score.

---

### Official Review · Reviewer_BxhQ · 2024-11-01

**Soundness:** 3
**Presentation:** 3
**Contribution:** 2
**Rating:** 6
**Confidence:** 2

**Summary:**

This article introduces JudgeBench, a novel benchmark for evaluating LLM judges, focusing specifically on their ability to assess factual and logical correctness in complex responses. They developed a pipeline to convert existing datasets with real labels into challenging response pairs, creating a benchmark of 350 pairs in the domains of knowledge, reasoning, mathematics, and coding. The key innovation is focusing on cases where even strong LLMs struggle to consistently generate correct answers, making the benchmark particularly challenging.

**Strengths:**

1. This paper aims to address the pressing need for a reliable method/benchmark to evaluate LLM judges as LLMs become more advanced and task complexity continues to scale up.
2. This work pinpoints a critical gap in existing benchmark frameworks for LLM judges by concentrating on factual/logical correctness instead of just instruction following ability or human preference alignment.
3. The proposed benchmark is significantly more challenging than existing benchmarks on LLM judges.
4. The paper proposes a novel pipeline to generate challenging response pairs tailored for LLM judges.

**Weaknesses:**

1. The benchmark only has 350 response pairs, and it's unclear if this size is sufficient to obtain robust conclusions about LLM judge performance. Will increasing the size to 500 or 1000 change the performance rank of different LLM judges and reward models?

2. Table 1 shows that the proposed benchmark is quite challenging, with 10 out of 14 evaluated models/judges performing far below 50% accuracy. It is counterintuitive to me that most evaluated models/judges perform much worse than random guessing (50% accuracy). Are there any superficial artifacts in this benchmark that might lead these LLM judges to incorrect answers? Please correct me if there is any misunderstanding.

3. The paper would benefit from more extensive validation to ensure that the selected response pairs truly represent challenging cases.

4. The key insight—"If a model struggles to consistently generate correct, coherent responses to a challenging question, it will find it difficult to differentiate between those responses"—utilized in this paper is not empirically verified.

**Questions:**

Please see the weakness section above.

Additionally:

When the data collected in this paper is released online, there is a possibility that it may be included in future LLM judge training, raising concerns about data contamination. This issue could impact the benchmark’s long-term effectiveness. Are there practices in place to prevent this and ensure the benchmark remains reliable over time?

---

> ### Author Response · Authors · 2024-11-18
> **To Reviewer BxhQ**
>
> We thank the reviewer for your valuable feedback.
>
> **On the size of JudgeBench.** Please see "Response to common comments" above.
>
> **On judges performing worse than random guessing.** Please see "Response to common comments" above.
>
> **The key insight not empirically verified.** Please see "Response to common comments" above.
>
> **On extensive validation.** We employ many strategies to ensure the quality of our pipeline represents truly challenging cases. First, as detailed in Section 3 (Data Filtering and Selection), we perform a meticulous filtering process. We utilize GPT-4o-mini as a solution verifier to remove responses that are incorrect solely due to formatting issues. Upon manual inspection, all filtered-out responses were confirmed to contain only formatting errors. For coding problems, incorrect responses are further validated using a coding checker to ensure errors are not due to formatting. This process guarantees that incorrect responses are genuinely incorrect and not artifacts of formatting.
>
> Second, we did extensive experiments to evaluate different LLM models on JudgeBench. The relative rank of the models are, in general, aligned with ranks from other reasoning benchmarks (e.g. GPQA, MATH), where larger models (e.g. Llama-3.1-70B-Instruct vs Llama-3.1-8B-Instruct) and models with stronger reasoning ability (e.g o1-preview, Claude-3.5-sonnet) in general performs better on this task. This indicates that our benchmark is not artificially designed to be challenging to models (for example, one could ask LLM to guess a random coin flip's result and all LLM's accuracy wouldn't surpass 50%), but indeed truly reflects the reasoning ability of the judges.
>
> Finally, our in-depth analysis of failure cases (see Appendix A.2) provides insights on why some finetuned judges underperform random guessing. See "**On failure case analysis**" from "Response to common comments" above for more details. These observations further validate the reliability of JudgeBench in assessing judges' reasoning ability.
>
> **On addressing data contamination.** Thank you for raising this important point! Our work indeed takes data contamination into account. The pipeline we propose in JudgeBench is very flexible and can be adapted to transform any challenging dataset with verification mechanism into challenging response pairs. In fact, many of the datasets we use in JudgeBench are already "live" (e.g., LiveBench, LiveCodeBench). These datasets are specifically designed with contamination in mind and are regularly updated to mitigate this issue. As LLMs continue to evolve, our pipeline can be applied to transform more challenging datasets (e.g., Olympiad Math) into this format, ensuring that the difficulty of our dataset evolves over time.

---

> > ### Author Response · Authors · 2024-11-23
> >
> > Dear reviewer BxhQ,
> >
> > Thank you for your thoughtful review and feedback on our paper. We hope our responses have adequately addressed your concerns.
> >
> > As the author-reviewer communication window is coming to an end, please let us know if you have any additional questions or comments. We would be happy to provide further clarification.
> >
> > We greatly appreciate the time and effort you have dedicated to reviewing our work.
> >
> >
> > Best regards, The JudgeBench Team

---

> > > ### Comment · Reviewer_BxhQ · 2024-11-24
> > >
> > > I thank the authors for the detailed responses, which have addressed many of my concerns. As a result, I have increased my score.

---

### Official Review · Reviewer_bwzk · 2024-11-02

**Soundness:** 3
**Presentation:** 3
**Contribution:** 2
**Rating:** 6
**Confidence:** 4

**Summary:**

- The paper proposes a new benchmark, *Judge Bench*, for evaluating large language models (LLMs) acting as judges, with a focus on knowledge, reasoning, math, and coding skills.
- Existing LLM-as-judge evaluations prioritize human preferences, assuming these preferences are reliable indicators of quality. However, this assumption can be weak for complex tasks, such as verifying mathematical proofs, where crowd-sourced annotators may lack the necessary domain expertise.
    - These existing benchmarks tend to emphasize stylistic preferences or instruction-following behaviors rather than logical accuracy and factual correctness.
- *Judge Bench* argues that effective evaluations should consider three core principles:
    - **Principle 1:** Responses must follow human instructions accurately.
    - **Principle 2:** Responses must be factually and logically correct.
    - **Principle 3:** Responses should stylistically align with human preferences.
- **Key contributions:**
    - A straightforward approach to generating evaluation data for LLM-as-judge models.
    - *Judge Bench*, a benchmark consisting of 360 challenging data points.

**Strengths:**

- Highlights the overemphasis on stylistic preferences in existing LLM-as-judge benchmarks, often at the expense of true task completion.
- Significant finding that many current LLM-as-judge models perform close to a random baseline on *Judge Bench*, revealing the difficulty of the prompts and the limitations of existing models.

**Weaknesses:**

- The benchmark includes only 360 examples, much smaller than many existing benchmarks such as Reward Bench.
- Beyond accuracy metrics, failure cases would add valuable insights into judge performance.
- Although the paper claims to apply a hierarchical approach, it appears that Principles 1 and 3 are largely overlooked in constructing *Judge Bench*.

**Questions:**

- It is interesting that the accuracy of judges drops when they are tasked with self-evaluation, which seems to contradict the prevailing research advocating for self-assessment or self-critique in LLMs. A deeper discussion on this aspect would be beneficial.

---

> ### Author Response · Authors · 2024-11-18
> **To Reviewer bwzk**
>
> Thank you for your valuable feedback.
>
> **On the dataset size** Please refer to "Response to common comments" above.
>
> **On failure cases.** Please refer to "Response to common comments" above.
>
> **On overlooking Principle 1 and 3.** While we do not cover stylistic preference (Principle 3), our work can be viewed as covering both Principle 1 and 2. This is because to answer the question correctly, the model has to first follow the instruction and answer that question, and failure to follow instruction would definitely result in a wrong answer. With that being said, we would like to highlight that the main emphasis of our work is indeed Principle 2. This is because Principle 1 and 3 has both been previously studied (e.g., LLMBar and MTBench), but Principle 2 is rarely looked at. Our work is complementary to these prior benchmarks and aim to fill the gap for Principle 2, which is missing in the current literature.
>
> **On accuracy drop for self-evaluation.** Thanks for raising this interesting point. LLM's ability to self-evaluate is an open research question, with some evidence suggesting they can [1,2] and other evidence suggesting they cannot without some form of external feedback (e.g., tools or oracle labels) [3,4]. Our results do not contradict the positive findings, since despite the observed drop in performance, LLMs in our study can still distinguish their own correct and incorrect responses a certain percentage of the time. This indicates that refinement based on self-evaluation is indeed possible. However, based on our results, leveraging another model with similar overall capabilities may prove to be a more effective approach for evaluation.
>
> [1] https://arxiv.org/abs/2303.17651
> [2] https://arxiv.org/abs/2305.11738
> [3] https://arxiv.org/abs/2402.11436
> [4] https://arxiv.org/abs/2310.01798

---

### Official Review · Reviewer_GJ2i · 2024-11-04

**Soundness:** 4
**Presentation:** 4
**Contribution:** 4
**Rating:** 8
**Confidence:** 3

**Summary:**

This paper introduces JudgeBench, a new benchmark designed to evaluate LLM-based judges on complex tasks across knowledge, reasoning, math, and coding domains. While LLM-based judges are widely used as scalable alternatives to human evaluation, their reliability is often taken for granted. JudgeBench addresses this gap by objectively evaluating LLM-based judges, particularly in tasks where human preference alone may not accurately indicate factual correctness. The benchmark employs a novel pipeline to convert challenging datasets into response pairs with preference labels for objective evaluation. Experiments show that JudgeBench is significantly more challenging than existing benchmarks, with many strong models performing only slightly better than random guessing, thus providing a reliable framework for assessing advanced LLM-based judges.

**Strengths:**

- This paper is well-written and well-motivated, with a strong emphasis on the motivation, which is crucial in the LLM-as-a-Judge domain.
- The paper proposes a new benchmark dataset for LLM-based judging, called JudgeBench, and demonstrates the effectiveness and challenging nature of their proposed dataset compared to other similar judging benchmarks, which is a significant contribution in the field of new dataset development.
- The authors conduct extensive experiments to demonstrate JudgeBench's effectiveness, testing various baseline methods and comparing it with different judging benchmarks. Additionally, they perform comparisons with reward models and RewardBench and investigate potential biases commonly recognized in the LLM-as-a-Judge research field.

**Weaknesses:**

- There is no technical novelty; however, considering that this paper proposes a novel benchmark dataset, this is acceptable, as the paper makes a significant contribution to this field.
- The authors conduct extensive experiments; however, I have a few suggestions/questions that do not impact the overall rating score:
  - The authors investigate various biases, but I am curious about length bias, a well-documented issue in this field where LLM-based judging models tend to prefer longer responses over shorter ones. My question is: when constructing the dataset, did the authors consider the length difference between "correct" and "incorrect" answers?
  - In Tables 1 and 2, the performance in the "Math" domain is relatively higher than in other domains (e.g., Knowledge, Coding). I am concerned that the "Math" domain samples in JudgeBench may be less challenging compared to other domains. Additionally, does the JudgeBench dataset include subcategories? For example, within "Math," are there various categories based on subjects (e.g., geometry), and within "Coding," are different programming languages and problem difficulties represented? If so, could the authors present detailed experimental results based on these finer-grained dataset samples?

**Questions:**

Please refer to Weaknesses.

---

> ### Author Response · Authors · 2024-11-18
> **To Reviewer GJ2i**
>
> Thank you for your valuable feedback.
>
> **On the lack of technical novelty.** A significant, but overlooked, contribution of this work is the novel pipeline used to generate the JudgeBench benchmark, which can transform any existing dataset for evaluating LLM performance into a dataset for evaluating the performance of LLM-based judges. This is critical, as there exist far fewer meta-evaluation benchmarks than standard LLM benchmarks, despite recent research supporting the importance of strong judges/verifiers for training more reliable models [1,2] and for effectively scaling inference-time compute [3,4].
>
> **On length bias.** Thanks for raising this point. Because each of our pairs contains two responses sampled from the same model, rather than from two different models, the responses tend to be of similar length. On average, across all instances of JudgeBench, correct and incorrect responses contain 562.29 and 561.16 tokens, respectively, using the GPT-4o tokenizer. This negligible difference demonstrates that the construction of JudgeBench effectively mitigates length bias, allowing LLM-based judges to be evaluated without this confounding factor.
>
> **On the strong performance on math.** This is a rather interesting observation! We sourced the math and reasoning questions from LiveBench, whose leaderboard [5] indicates that LLMs tend to perform better on the reasoning category than the math category. We observed this too, with 3 of the 4 models in Table 5 achieveing higher accuracy on solving reasoning questions than math questions. Yet, when it comes to judging, these same models tend to exhibit higher accuracy on the math pairs than the reasoning pairs (see Table 5). Given this, we hypothesize that the strong performance we observed on the math subset of JudgeBench is because LLM-based judges are better equipped at verifying responses to math questions than other types of questions (e.g., reasoning), and not because the math questions themselves are less challenging.
>
> **On the granularity of the results.** The existance of subcategories within JudgeBench is dependent on the source dataset. For example, the MMLU-Pro dataset from which we derive the knowledge category contains 14 subcategories (e.g., Law, Engineering, and Economics). Though JudgeBench includes an equal number of pairs derived from each subcategory, there are likely not enough pairs per subcategory to draw robust conclusions. Other source datasets include no natural subsets, e.g., our code category is derived from LiveCodeBench which is entirely in python.
>
> [1] https://arxiv.org/abs/2407.13692
> [2] https://arxiv.org/abs/2305.20050
> [3] https://arxiv.org/abs/2408.15240
> [4] https://arxiv.org/abs/2407.21787
> [5] https://livebench.ai/

---

> > ### Comment · Reviewer_GJ2i · 2024-11-26
> > **Response by Reviewer**
> >
> > Thank you for clearly addressing my questions.

---

### Author Response · Authors · 2024-11-18
**To All Reviewers**

We sincerely thank the reviewers and organizing committee for their thoughtful feedback and efforts in evaluating our work. We are grateful for the recognition of our contributions and would like to highlight some positive comments:

* The paper is well-motivated and addresses an important and urgent problem in the domain of LLM-based judges (GJ2i, BxhQ, VLGm).
* Our proposed pipeline for generating challenging response pairs is recognized as novel (GJ2i, BxhQ).
* Our extensive experiments effectively demonstrate the utility and robustness of the proposed benchmark (GJ2i, BxhQ).
* Our experiments reveal intriguing insights and significant findings in the evaluation of LLM-based judges (Bwzk, VLGm).
* The paper is well-written, with all reviewers rating the presentation above 3.

In the remaining space, we address comments raised by multiple reviewers. Individual comments and suggestions are answered in detail in the respective sections below. Once again, we express our gratitude to the reviewers for their valuable input.

---

> ### Author Response · Authors · 2024-11-18
> **Response to common comments**
>
> **On the size of JudgeBench (bwzk, BxhQ, VLGm).** First, JudgeBench is of similar size to related benchmarks. For instance, FairEval [1] contains 80 unique questions, LLMEval-2 [2] contains 480, MT-Bench [3] contains 80, and LLMBar [4] contains 419. RewardBench [5] is larger, but it's an aggregration of existing benchmarks, including MT-Bench and LLMBar.
>
> Importantly, JudgeBench novel pipeline enables us to transform any existing dataset for evaluating LLM performance into a dataset for evaluating the performance of LLM-based judges. To demonstrate the flexability of our pipeline and the robustness of the results, we augmented our "knowledge" subset, increasing the number of response pairs from 154 to 770. We evaluated several LLMs using the Arena-Hard prompt, and observed that the relative rankings among these judges were the same between our original set and the augmented set, despite small variations in the scores themselves (see the table below).
>
> | Model                    | Original Set | Augmented Set |
> |--------------------------|--------------|---------------|
> | gpt-4o                   | 50.65 _(3rd)_| 46.49 _(3rd)_ |
> | gpt-4o-mini              | 48.05 _(4th)_| 44.03 _(4th)_ |
> | claude-3.5-sonnet        | 62.34 _(1st)_| 63.25 _(1st)_ |
> | claude-3-haiku           | 35.06 _(6th)_| 39.35 _(6th)_ |
> | llama-3.1-70b-instruct   | 51.30 _(2nd)_| 52.60 _(2nd)_ |
> | llama-3.1-8b-instruct    | 38.31 _(5th)_| 40.00 _(5th)_ |
>
> [1] https://arxiv.org/abs/2305.17926
> [2] https://arxiv.org/abs/2312.07398
> [3] https://arxiv.org/abs/2306.05685
> [4] https://arxiv.org/abs/2310.07641
> [5] https://arxiv.org/abs/2403.13787
>
> **On judges performing worse than random guessing (BxhQ, VLGm).** We explain our evaluation method in Section 4 (L286). Each response pair is evaluated **twice**, swapping the order of the pairs in the second trial. Judges' decisions are aggregated across both trials. For instance, if the decisions are (A1 > B1 and A2 = B2), the final aggregate decision is A > B. This final aggregate decision is what we use to compute the judge's accuracy on JudgeBench. Inconsistent decisions (e.g., A1 > B1 and A2 < B2) are interpreted as an aggregate of A = B and deemed incorrect.
>
> This evaluation method is specifically designed to penalize random guessing. If a judge cannot produce consistent responses across trials, it indicates an inability to distinguish between the response pairs, equivalent to random guessing, which is marked as incorrect. Appendix B.2 (Tables 5 and 6) provides detailed statistics on fine-tuned judges' decisions, revealing a high ratio of inconsistent decisions as a major contributor to poor performance.
>
> We would also like to highlight that, when comparing JudgeBench to other benchmarks, we apply the same evaluation methods to all benchmarks, so our benchmark does not gain any advantage from the evaluation method. Under this consistent setup, JudgeBench demonstrates higher challenge and separability than prior benchmarks.
>
> **On failure cases (bwzk, VLGm).** Our work does provide failure case analysis, which is detailed in Appendix A.2 and referenced in Section 4.2 (L397). In A.2, we provide detailed statistic of fine-tuned judges decisions, and an in-depth analysis on their poor performance. One main reason for the poor performance is inconsistency between judgements. Table 6 shows that fine-tuned judges that fall below the random baseline all have pretty high ratio of inconsistent judgements (29.14%-59.71%), this indicates that they are incapable of distinguishing the correct response and are just doing random guessing. Other than inconsistent judgements, some judges (e.g. Prometheus2-bgb-8x7b, PandaLM-7B) frequently output invalid decisions (e.g. Neither A or B, 10/10, 3), which also contributes to the high error rate.
>
> **The key insight not empirically verified (BxhQ,VLGm).** Our main insight posits: "If a model fails to generate correct, coherent responses, it will inherently struggle to distinguish between **its own responses** as a judge." This is empirically verified in Section 4.4 (Figure 4). We use Claude-3.5-Sonnet as the base model to construct datasets and evaluate accuracy across five representative LLMs. As predicted, Claude-3.5-Sonnet finds its own generated response pairs particularly challenging, with accuracy dropping significantly from 64.3% to 44.8%, ranking below GPT-4o and Llama-3.1-70B-Instruct. Importantly, the relative ranking of other models remains unchanged.
>
> Meanwhile, the dataset's difficulty is transferable across models. When a stronger reasoning model (Claude-3.5-Sonnet) generates response pairs, they become more challenging for other models. All four other models show accuracy drops relative to GPT-4o pairs. This validates our insight and further demonstrates the robustness of JudgeBench.

---

### Meta-Review · Area_Chair_XVGZ · 2024-12-13

**Metareview:**

The article introduces JudgeBench, a novel benchmark designed to evaluate the performance of LLM judges in assessing factual and logical correctness in complex responses. This benchmark features a pipeline that transforms existing datasets with real labels into challenging response pairs, resulting in a collection of 350 pairs across knowledge, reasoning, mathematics, and coding domains. The key innovation of JudgeBench lies in its focus on scenarios where even advanced LLMs struggle to produce consistently correct answers, thereby creating a more rigorous testing environment for LLM judges. This makes JudgeBench particularly valuable for advancing the evaluation of AI models as they become increasingly sophisticated.

**Additional Comments On Reviewer Discussion:**

All reviewers have reached a consensus that this paper is strong and well-organized. The authors should revise the paper as mentioned in the reviews.

---

### Decision · Program_Chairs · 2025-01-22

Accept (Poster)